# Evaluation of the HEAL™ing Mental Health program: A prospective cohort study of short-term changes from a physical activity and lifestyle education program for people with mental health disorders living in rural Australia

**Annette J. Raynor** [1,2]*, **Sophia Nimphius** [1,2], **Daniel Kadlec** [1,2], **Sally Casson** [1,2], **Caitlin Fox-Harding** [1,2], **Lauren V. Fortington** [1]

1 School of Medical and Health Sciences, Edith Cowan University, Joondalup, Western Australia, Australia,
2 Centre for Human Performance, Edith Cowan University, Joondalup, Western Australia, Australia

* a.raynor@ecu.edu.au

## Abstract

This study aimed to evaluate short-term outcomes of the HEAL™ing Mental Health program, an 8-week intervention for change in functional, behavioural and physiological health and wellbeing designed for people living with mental health conditions in rural or regional areas of Australia. A prospective cohort study was completed, reporting on 19 items (pre-program) and 15 (post-participation change), organised across seven domains. Participants took part in an Accredited Exercise Physiologist/Nurse led supervised group exercise (60 minutes) and healthy lifestyle education program (60 minutes). Separate linear mixed models with restricted maximum likelihood were used to examine the primary research question considering the effect of the program on: walking (min/week); planned, incidental and total physical activity (min/week); sitting time; active days; fruit and vegetable intake; body mass index; waist circumference; blood pressure; 6 minute walk distance; 30 second sit-to-stand; psychological distress symptoms; and stage of behaviour change. There were 99 participants (31 males, 68 females) out of 117 participants completed more than 50% of program sessions. Twelve of 15 measures achieved their desired target change and a statistically significant change toward the desired outcome was reported for 14 of 15 measures. Positive results were obtained for participants completing more than 50% of sessions, suggesting that HEAL™ ing Mental Health program is effective to increase physical activity and healthy lifestyle choices in individuals who self-report a mental health disorder.

## Introduction

More than one quarter (28%) of the Australian population live in rural or remote areas [1], benefiting from many positive aspects including strong social cohesiveness, better work-life

**Data Availability Statement:** Data cannot be shared publicly because of privacy restrictions. To request access to the data, please contact Exercise and Sports Science Australia via info@essa.com. au.

**Funding:** Exercise & Sports Science Australia (ESSA) coordinated the data collection but had no role in the analysis or its reporting.

**Competing interests:** The authors have declared that no competing interests exist.

balance and shorter commutes to work [2–6]. However, the smaller population centres combined with long distance from major cities, also mean that rural and remote populations can experience disadvantages with some socio-economic indicators such as lower education, unemployment, and limited access to healthcare. Health promoting interventions that work around known disadvantages are important for equitable health opportunities.

Compared to metropolitan populations, people living outside of major Australian cities experience poorer physical and mental health outcomes, higher levels of disease and injury, and earlier mortality [1]. Several factors contribute to these differences in health. As examples, people living in rural and remote areas worldwide face barriers with the built environment for participation in physical activities [7] and socioeconomic disadvantage that makes healthy food less affordable [8]. The impact of these health determinants is seen through different measures of diet and physical activity. National health survey data shows that only 10% of Australians living outside major cities meet dietary recommendations for vegetable intake and fewer than 50 percent meet fruit intake recommendations [1]. Further, rural populations do not meet Australian physical activity and exercise guidelines, with physical activity levels lower than those of urban populations [9].

Mental health in rural populations is also influenced by socioeconomic factors that tend to be amplified by distance from metropolitan centres, including a person's access to psychological services, living conditions and employment status [10]. People living with mental health conditions are more likely to develop comorbid physical illness and have a reduction in life expectancy compared with the general population [11]. Although modifiable lifestyle behaviours such as increasing physical activity and improving diet are known to influence the incidence of chronic diseases and life expectancy [12, 13], people with mental health conditions experience barriers to improving these behaviours, such as high levels of perceived stress, a lack of self-confidence and limited social support [14].

Interventions that support sustainable changes in lifestyle, addressing diet, exercise and wellbeing have been shown as beneficial to improve general health in regional and remote areas. The Healthy Eating Activity and Lifestyle (HEAL™) program is an evidence-based lifestyle program that incorporates physical activity and diet and has been proven effective at inducing weight loss and improving lifestyle behaviours in adults living in rural areas of Australia [15]. The program demonstrates that a community-based physical activity and lifestyle educational program is effective at improving a range of health measures including increased physical activity, increased fruit and vegetable consumption, improved functional capacity measures, improved blood pressure and reduced weight and waist circumference. Evaluations of the HEAL™ program have been in urban, regional and remote populations, and have involved diverse populations including Aboriginal and Torres Strait Islanders, and people from refugee and migrant backgrounds [16, 17]. An important, yet, to date, not investigated population for the HEAL™ program are individuals with mental health disorders.

In previous programs, HEAL™ did not include measures of current or changed mental health symptoms. The base-program has since evolved to offer HEAL™ing Mental Health with some content, language and overall tone being modified to reflect the intended population. To determine the impact of these modifications, this study aimed to evaluate change in physical, functional and behavioural factors from baseline to completion of the HEAL™ing Mental Health program in rural Australian populations with self-reported mental health distress.

## Methods

This study is a prospective cohort study, reporting pre- and post-intervention change in seven domains. Data were collected by the site facilitator and entered into individually coded

spreadsheets which were sent to the HEAL™ project team. The data was then provided in an anonymised format (re-identifiable to Exercise & Sports Science Australia (ESSA)) to the research team for analysis and reporting. The study details were submitted for ethics review and determined to be exempt from Human Research Ethics Review [2022-03507-RAYNOR].

The intent was to recruit individuals who self-reported as living with a mental health condition from 21 rural and remote regions across Australia, which included communities with Culturally and Linguistically Diverse (CALD), lower socio-economic and Aboriginal and Torres Strait Islander populations. A total of 182 participants were recruited to participate in the HEAL™ing Mental Health Program, including 117 in the adult mental health group, 32 completing a standard adult HEAL™ program and 33 in a teen mental health program. No individuals identified as being part of the CALD or an Aboriginal or Torres Strait Islander community and it was assumed that all participants were English first language people. This study is focused on the 117 participants in the adult mental health group.

The program was delivered between May and October 2022 at eighteen centres staffed by Accredited Exercise Physiologists (AEPs) (providing instruction to 72 of the 99 completing participants) and one centre where the program was facilitated by a registered nurse (27 participants). All HEAL™ facilitators had undertaken the HEAL™ facilitator training as provided by ESSA and completed a competency-based assessment and written theory assessment prior to certification. The HEAL™ facilitators are trained and encouraged to use principles of self-managed change in the delivery of their programs. Facilitators are responsible for pre-exercise screening and initial assessment, delivery of the nutrition, physical activity and behaviour modification education sessions, group exercise sessions, post program assessment and a follow-up review.

The HEAL™ing Mental Health program is a modified version of the original HEAL™ program, which was first developed by a team of exercise physiologists, dietitians and chronic disease specialists at South Western Sydney Medicare Local [15]. The standard program is 8 weeks in duration and includes one hour of supervised group exercise per week, followed by a one-hour lifestyle education session. The 60-minute supervised exercise session included a 5-minute warm up and cool down, with 45–50 minutes low-moderate intensity exercise, using a circuit style delivery mode with stations of upper and lower body resistance and aerobic exercises. The educational content of the adult mental health program was similar to the standard program with one new topic, the benefits and barriers of physical activity being introduced, and the 'Nutrients in your diet' session being combined with 'What is healthy eating' session. When delivering the mental health version of the educational sessions, some potential anxiety provoking triggers were removed, with some materials being simplified or removed. The tone of the information was also modified in parts to be more suggestive than consequential with additional detail provided on the negative effects of alcohol consumption and the measurement of standard drinks (Table 1).

Measures collected for the HEAL™ing mental health include 19 individual items organised across seven domains (Table 2). Data were collected at two timepoints (pre and post) completion of the 8-week program, with HEAL facilitators following specific protocols for collection. Fifteen of the 19 items are reported as measures of change pre- and post- intervention.

Functional measures in Domain 1 focus on physical activity (PA) with questions drawn from questionnaires used in previous research, and originally from the Active Australia Survey [18]. Specific measures include the weekly minutes spent walking for more than 10 minutes and completing other activities, such as gardening or household chores, where participants are required to breathe hard. The three individual PA variables were summed to provide the total PA reported as minutes per week. Sitting time was captured using a question from the

**Table 1. Content of the eight weekly education sessions delivered through the HEAL™ing Mental Health program.**

|  | Educational topic |
|---|---|
| Session 1 | Your health and your choices |
| Session 2 | Physical activity |
| Session 3 | Physical activity: benefits and barriers |
| Session 4 | What is healthy eating? Including nutrients in your diet |
| Session 5 | Meal planning and eating on the move |
| Session 6 | Label reading |
| Session 7 | Making and maintaining a healthy lifestyle |
| Session 8 | Myths and misconceptions and non-hungry eating |

**Table 2. Domains and specific data collected, noting desired target for change with participation.**

| OUTCOME MEASURE/TOOLS | SPECIFIC VARIABLE DATA COLLECTED (n = 19 items) | TARGET CHANGE FOR PARTICIPANTS |
|---|---|---|
| **DOMAIN 1: FUNCTIONAL MEASURE** |  |  |
| Adult pre and post screening questionnaire developed by ESSA<br>Pre and post physical activity functional measures and questionnaire* | 1. Walking (min/week)<br>2. Planned PA (min/week)<br>3. Incidental PA (min/week)<br>4. Total PA (min/week) ^<br>5. Days activity (d/week)<br>6. Sitting (h/d) | Improve physical activity levels by a minimum of 10 minutes per week<br>Decrease time spent sitting by 30 minutes per day |
| **DOMAIN 2: BEHAVIOURAL** |  |  |
| Pre and post program fruit and vegetable functional measures and questionnaire* | 7. Fruit intake (serves per day)<br>8. Vegetable intake (serves per day) | Increase total serves fruit and vegetables by half a serve per day |
| **DOMAIN 3: PHYSIOLOGICAL HEALTH MEASURES** |  |  |
| Pre and post program physiological and behavioural measures and questionnaire* | 9. Mass ^<br>10. Height ^<br>11. BMI<br>12. Waist cm<br>13. Diastolic blood pressure<br>14. Systolic blood pressure | Mass—used to calculate BMI, baseline only (BMI used in post-test)<br>Height—baseline only, no change<br>BMI—move closer to a healthy BMI<br>Waist—reduction in circumference cm<br>Diastolic blood pressure—decrease resting BP by 2 mmHg<br>Systolic blood pressure—decrease resting BP by 2 mmHg |
| **DOMAIN 4: CARDIOVASCULAR FITNESS** |  |  |
| 6-minute Walk Test | 15. Distance walked in 6 minutes | Increase distance by 25 m |
| **DOMAIN 5: LOWER BODY STRENGTH** |  |  |
| 30-sec Sit to Stand Test | 16. Number of sit to stand | Increase number by 5 |
| **DOMAIN 6: PSYCHOLOGICAL DISTRESS** |  |  |
| K10 Questionnaire (Black Dog Institute) | 17. Score on K10 self-report | Reduce K10 score |
| **DOMAIN 7: READINESS FOR CHANGE & PARTICIPATION** |  |  |
| Stages of Behaviour Change Tool<br>Session attendance | 18. Stage of change<br>19. Facilitator observation and headcount of attendance /compliance to program sessions ^ | Increase participation and participants readiness for change (exercise levels and mental illness)–improve the stage of change by one level<br>Results focused on participants with >50% attendance (8 of 16 sessions) |

*available from Exercise & Sports Science Australia (ESSA)

^variable not reported for change over time

International Physical Activity Questionnaire (IPAQ) on the average number of hours per day sitting [19].

Fruit and vegetable intake are the focus of Domain 2, measured as self-reported average intake per day. Cues are provided to participants to assist with understanding what a serving size looks like for different fruit/vegetables. Domains 4 and 5 use common measures of 30 second sit-to-stand and 6-minute walking distance to ascertain lower body strength and cardiovascular fitness respectively. The sit-to-stand has been found to have excellent test-retest reliability (r = 0.89, 95% CI 0.79–0.93) and intra and inter-rater reliability (r = 0.95, 95% CI, 0.84–0.97) [20]. The 6-minute walk test is an easy to administer field test to assess functional capacity in healthy adults and can also be used with those who have chronic conditions or as an outcome measure during cardiovascular or pulmonary rehabilitation. Test–retest reliability has been assessed in several different populations including older adults with excellent test-retest reliability (ICC of 0.95) [21] and good concurrent validity with the sit to stand (r = 0.67) [22].

Domain 6 reports on the Kessler Psychological Distress Scale (K10), a self-report indicator for mental distress commonly used in general population surveys [23]. Participants report against 10 symptoms of anxiety and depression experienced in the previous four weeks, resulting in a score between 10 and 50. Scores are reported as mean and standard deviation, and can be categorised into groups [24] of:

- Scores 10–19: likely to be well (no indication of a mental health disorder)

- Scores 20–24: likely to have a *mild* mental health disorder

- Scores 25–29: likely to have a *moderate* mental health disorder

- Scores 30+: likely to have a *severe* mental health disorder.

Domain 7 looked at participation in the sessions (simply, attendance of participants, reported by facilitator) and The Stages of Behaviour Change which provides an indication of the individuals readiness to participate in an exercise program, with five possible stages from pre-contemplation to maintenance [25].

Data were received as one excel spreadsheet, with information provided for participant identifier, the site of program delivery, program attendance (total number) and pre- and post-results for each of the 19 measurements. Data were reviewed, restructured, visualised, and summarised to detect and correct any potential errors or implausible data points, prior to performing the statistical analysis (e.g., height in cm was converted to height in metres).

Attendance was monitored each week by the site facilitator. Participants who did not attend more than 50% of the sessions were excluded from the main analysis. All but two of the excluded participants had dropped out before completing the post-testing.

Separate linear mixed models (LMM) with restricted maximum likelihood (REML) were used to examine the primary research question considering the effect of the program on the different measures, with time as a fixed effect and participant as a random effect. We chose REML as the way participants differ from one another may be related to their starting values. The REML enables more accurate estimates of the variance and model parameters when such correlation between fixed effects and random effects may be present. As only two time points were present, only random intercepts for participants were included as to not risk over fitting by also including random slopes. The degree of correlation between variables was examined prior to performing the separate LMMs. For variables with very large correlations (e.g., mass and BMI), only one variable was statistically examined to reduce repeated statistical tests that inflates the risk of a Type I error. Assumption of linearity, normality and homoscedasticity

were confirmed through inspection of plots. The effect of the intervention on stage of change was examined using a Wilcoxon Signed Rank test. Cohen's d effect size with 95% confidence interval was calculated for all variables using the pooled standard deviation. Statistical significance was set at an alpha level of 0.05 for all tests. All analyses were performed in R version 4.0.2 and, for transparency, all code used for the analysis is available from the authors, on request.

## Results

There were 117 participants from which 99 were included in analysis (31 males, 68 females) having completed more than 50% of program sessions. Included participants had a mean age of 59.5 years and mean BMI of 33.3 (Table 3). The 18 excluded participants had a younger mean age (53.3) and a higher BMI (35.5).

The mean K-10 score for included participants was 20.4 at the pre-program assessment with 53 participants classified as likely to be well, 21 likely to have a mild mental disorder, 8 likely to have a moderate mental disorder and 15 likely to have a severe mental disorder. The distribution for excluded participants across the K-10 groups was similar (S1 Table). Post-intervention testing showed a significant shift with a mean score of 17.2, moving from a moderate to a low level of psychological distress.

Prior to the program most participants were in the preparation and action phases of the Readiness to Change scale (Fig 1 and S1 Table). Excluded participants were represented similarly across all stages of readiness except pre-contemplation. Following the program there was a significant shift with most included participants now in action and maintenance phases.

In terms of changes from pre- to post-program, the results indicated a statistically significant change toward desired outcome for fourteen of fifteen measures. Twelve of these measures recorded the desired target change at the population (grouped) level (Fig 2, Table 4). The only measure not to record a significant change was diastolic blood pressure, though it shifted in a desired direction, and systolic blood pressure decreased as intended. There was a small decrease in BMI, whilst fruit and vegetable consumption increased but only marginally.

The frequency and volume of physical activity increased from pre- to post-testing, with the average time spent walking increasing by 54.8 min/week, with sitting time decreasing by 1.11 hours /day. Both the intent to be more physically active and the actual amount of physical activity completed improved at post-program testing (11.2 min/week increase and 26.1 min/week increase).

Significant improvement was seen in the cardiovascular and leg strength measures with the average distance covered during the 6-minute walk increasing by 43 m, whilst the number of sit to stand completions increased by 2.7.

**Table 3. Descriptive characteristics of participants at baseline (n = 99).**

|  | Attendance (sessions/16) | | Age (y) | | Height (m) | | Mass (kg) | | BMI (kg·m⁻²) | |
|---|---|---|---|---|---|---|---|---|---|---|
|  | mean | SD | mean | SD | mean | SD | mean | SD | mean | SD |
| Male (n = 31) | 13.84 | 1.68 | 60.6 | 16.3 | 1.85 | 0.07 | 101.9 | 24.3 | 32.3 | 7.15 |
| Female (n = 68) | 13.18 | 2.29 | 59.0 | 14.7 | 1.63 | 0.08 | 89.7 | 28.8 | 33.8 | 9.98 |
| Total (n = 99) | 13.40 | 2.13 | 59.5 | 15.2 | 1.67 | 0.09 | 93.7 | 27.9 | 33.3 | 9.14 |
| Excluded (n = 18, 16f, 2m) | < 8 | | 53.3 | 15.7 | 1.67 | 0.09 | 93.1 | 24.0 | 35.5 | 10.1 |

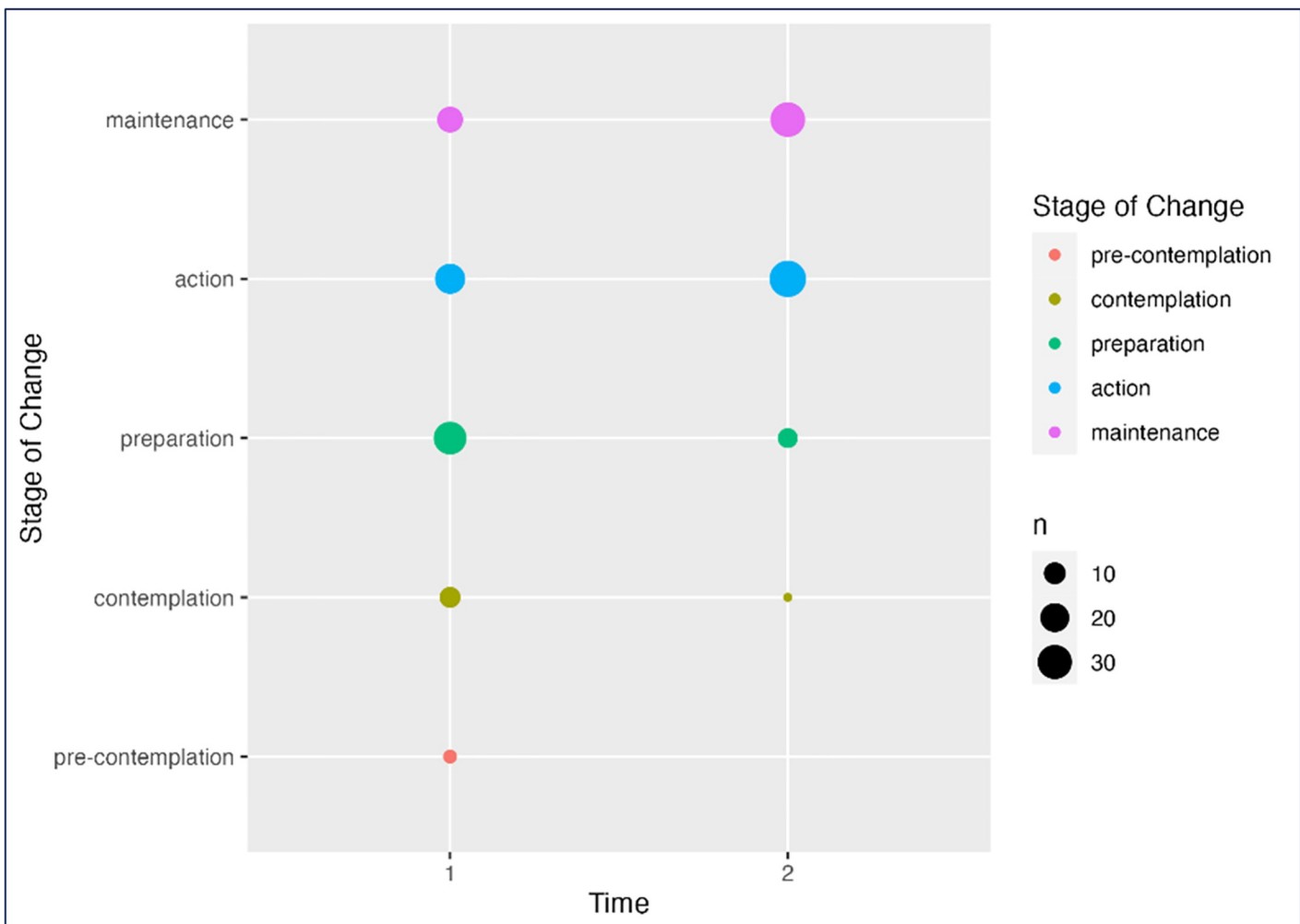

**Fig 1. Scores recorded for readiness to change—Pre and post program.**

## Discussion

This study reports on the short-term outcomes of the HEAL™ing Mental Health program, an 8-week intervention focused on functional, behavioural, and physiological health and wellbeing for people living with mental health conditions in rural or regional areas. Results of pre- and post- program tests showed a strong positive effect on participants' readiness to change, level of physical activity and mental wellbeing.

There are numerous studies reporting physical activity programs, consistently reporting positive effects of interventions, in different age groups and with different physical and mental health conditions [26–29]. Many of these intervention studies promoted similar program structures as the HEAL™ing Mental Health program which comprised of education and health promotion, together with a direct physical activity intervention. In their findings, focused on populations aged 60+, Taylor et al. highlight the notable lack of research in diverse and disadvantaged samples and social participation outcomes [28]. This is a gap the current study intended to fill. Unfortunately, despite the final sample including individuals from several small and medium sized rural towns, there were no participants who identified as Aboriginal

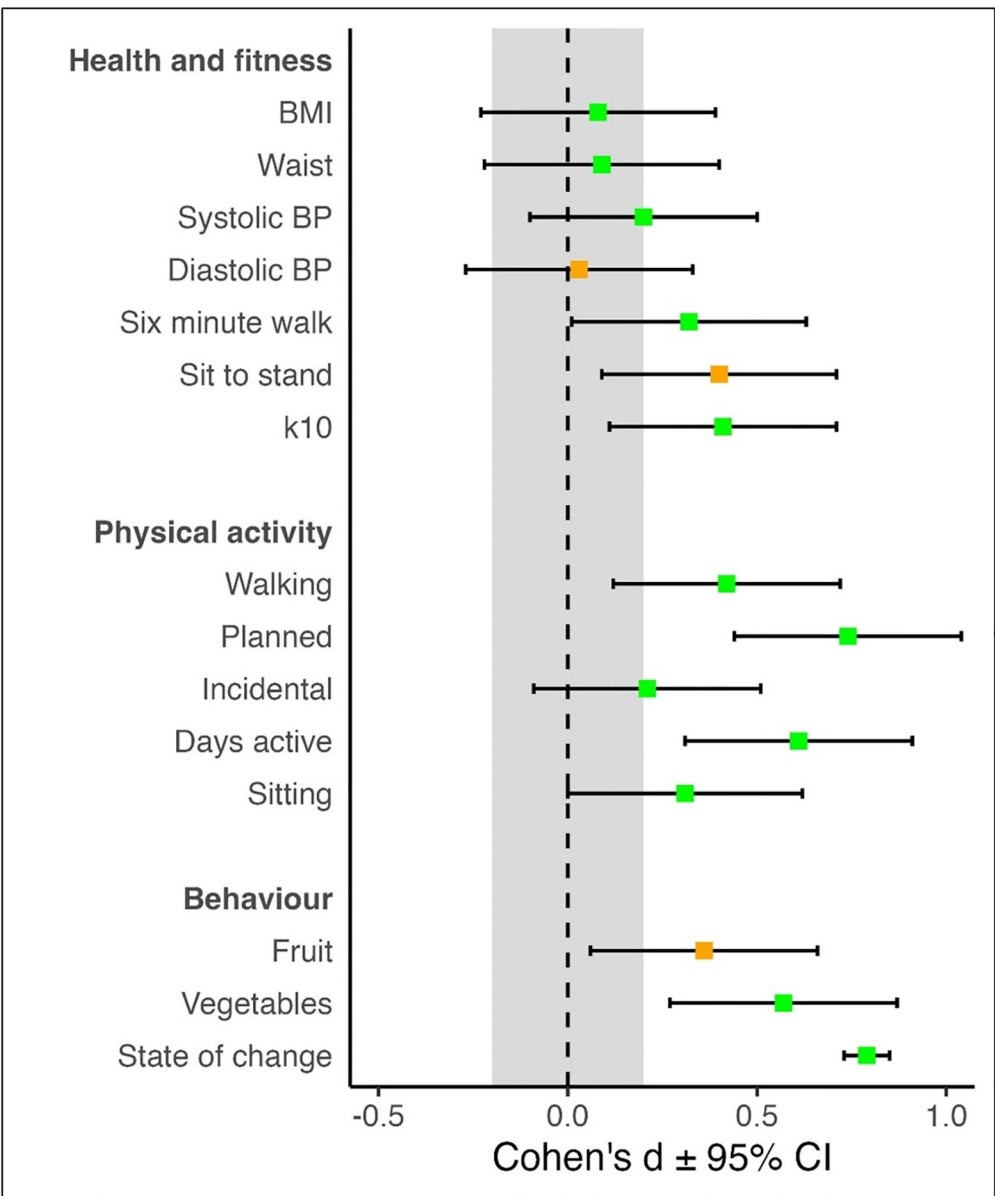

**Fig 2. Change in outcome measures pre- to post-program, with 95% CI (refers to data in Table 4).** Green
square = improved value and met pre-specified target; Orange square = improved value, but did not meet pre-specified
target; Refer to Table 2 for details of the measures and the desired change.

or Torres Strait Islander or being from a CALD community. Further study to identify the bar-
riers to recruitment of diverse populations to a free program such as the HEAL™ing Mental
Health program is a necessary next step. What is evident is the positive effect the program had
on those who were involved and who attended more than 50% of the sessions, and therefore
the benefits to a wider more disadvantaged population warrants further investigation.

All participants enrolling in the HEAL™ing Mental Health program had self-reported as
experiencing symptoms of mental health distress. The post-participation change showed
promising results with a decrease of over 3 points on the K-10. However, looking at the cohort,
on enrolment only 44 of the 97 participants were classified as *likely to have a mental health*

**Table 4. Changes in dependent variables (mean ± standard deviation) with effect size differences (Cohen's d ± 95% confidence intervals for participants with an attendance greater than 50%.**

| Variable | Pre | | | Post | | | Cohen's | | | |
|---|---|---|---|---|---|---|---|---|---|---|
| | n | mean | SD | n | mean | SD | *actual group change* | p | d | 95% CI |
| **DOMAIN 1: FUNCTIONAL MEASURES**<br>Walking (min/week) | 96 | 81.2 | 111 | 80 | 136 | 150 | +54.8 | < 0.01* | 0.42 | 0.12, 0.72 |
| Planned (min/week) | 97 | 65.6 | 74.7 | 80 | 121 | 76.8 | +46.3 | < 0.01* | 0.74 | 0.43, 1.04 |
| Incidental (min/week) | 97 | 88.9 | 137 | 80 | 115 | 106 | +26.1 | < 0.01* | 0.21 | 0.09, 0.51 |
| Days activity (d/week) | 96 | 3.89 | 2.31 | 80 | 5.12 | 1.63 | +1.23 | < 0.01* | 0.61 | 0.31, 0.91 |
| Sitting (h/d) | 91 | 6.58 | 3.94 | 76 | 5.47 | 3.04 | -1.11 | < 0.01* | 0.31 | 0, 0.62 |
| **DOMAIN 2: BEHAVIOURAL**<br>Fruit (n/d) | 97 | 1.54 | 0.94 | 79 | 1.87 | 0.91 | +0.33 | < 0.01* | 0.36 | 0.06, 0.66 |
| Vegetables (n/d) | 97 | 2.62 | 1.33 | 79 | 3.37 | 1.3 | +0.75 | < 0.01* | 0.57 | 0.26, 0.87 |
| **DOMAIN 3: PHYSIOLOGICAL HEALTH MEASURES**<br>Mass (kg) | 94 | 93.7 | 27.9 | 78 | 93 | 28.2 | -0.7kg | (reported within BMI) | | |
| BMI (kg·m$^{-2}$) | 92 | 33.3 | 9.14 | 71 | 32.5 | 9.23 | -0.8kg·m$^{-2}$ | < 0.01* | 0.08 | -0.23, 0.39 |
| Waist (cm) | 87 | 111 | 19 | 76 | 109 | 19.1 | -2.0cm | < 0.01* | 0.09 | -0.22, 0.40 |
| Hip (cm) | 85 | 115 | 19.8 | 75 | 115 | 21.2 | 0cm | No change | | |
| Systolic BP (mm/Hg) | 91 | 134 | 15.7 | 78 | 131 | 14 | -3.0mm/Hg | 0.02* | 0.20 | -0.10, 0.50 |
| Diastolic BP (mm/Hg) | 91 | 80.4 | 11.1 | 78 | 80.1 | 8.95 | -0.3mm/Hg | 0.79 | 0.03 | -0.27, 0.33 |
| **DOMAIN 4: CARDIOVASCULAR FITNESS**<br>Six-minute walk (m) | 90 | 401 | 131 | 76 | 444 | 132 | +43m | < 0.01* | 0.32 | 0.02, 0.63 |
| **DOMAIN 5: LOWER BODY STRENGTH**<br>Sit to stand (n) | 90 | 12.6 | 5.4 | 78 | 15.3 | 7.6 | +2.7 | < 0.01* | 0.40 | 0.09, -0.71 |
| **DOMAIN 6: PSYCHOLOGICAL DISTRESS**<br>k10 (score) | 97 | 20.4 | 8.4 | 77 | 17.2 | 6.9 | -3.2 | < 0.01* | 0.41 | 0.10, 0.71 |

* Indicates a significant difference between the conditions (p < 0.05). BP = Blood pressure.

*disorder* (of varying significance from mild, moderate and severe). Psychological distress was assessed using the K-10, a self-report tool with respect to symptoms over the past four weeks. The cohort's mean K-10 score at baseline (pre-intervention) was 20.4, suggesting data were skewed toward a more severe level, given Australian population norm values from 2007, which have a mean of 13.9 for adults of a similar age (55–64 years). Thus, we had a small group of high scoring participants, but the majority were in the 'lower' range. While there are slight variations in the way classification of K-10 scores can be applied, this impacts the lower-middle range of the scale (less severe symptoms) [30, 31]. Using an alternative scoring, [31] saw a small number of participants (n = 11) who scored 20 or 21 being classed as 'moderate' rather than 'mild', and an additional 10 participants being classed as 'high' rather than 'moderate' [24]. Critically, however, more than half of the participants were considered low to moderate severity when using either scoring system. Thus, the majority of this cohort have only minor levels of psychological distress which should be noted when findings are generalised to other populations. We did not look at outcomes by subgroup of K-10 severity scores as we had too few participants for meaningful outcomes but could be valuable in the future with larger numbers. This suggests the acute benefits of the program may be helping with symptoms of mental distress; however, a longer term follow up is important to determine the sustainability of this change.

The participants readiness to engage in lifestyle changes by way of physical activity and nutritional improvements was measured using The Stages of Behaviour Change tool, a

construct within the Transtheoretical Model [25]. This theoretical framework is an effective means to encourage behaviour change across a range of issues, including smoking, alcohol abuse, weight control and exercise [32]. The 5 stages of change, ranging from pre-contemplation to maintenance, depicts the dynamic evolution of behaviour change and the possibility of recycling through stages, encouraging non-punitive discussions to be had around relapses. This tool can help allied health and exercise professionals tailor an intervention to an individual's needs (e.g., those in the 'contemplation' stage may benefit from more education than those in the 'action' phase). Additionally, movement between stages has been shown to be a predictor of future readiness for change. Studies in addictive behaviours show that moving from one stage to the next within one month will double one's chances of acting on behaviour change in the next six months of treatment [33]. The targeted change for participants in the HEAL™ing Mental Health program was to improve the stage of change by one level, which was achieved with most included participants in action and maintenance phases by the end of the program. Given most of the current population were initially in the pre-contemplation, contemplation or preparation stage at the commencement of the program, and considered likely to moderate levels of psychological distress, the shift of many individuals to a stage of *action* or *maintenance* was a key indicator of the success of this program.

The improvements in the functional measures of increased physical activity (both planned and completed), the improved leg strength and increased cardiovascular fitness all serve to improve the capacity of the individual to participate more fully in activities of daily living, however they all remain below the expected levels [34, 35]. Similar levels of performance and change were seen in these functional measures when the standard versions of the HEAL™ program was delivered [15]. The novel inclusion of a healthy eating element and lifestyle educational sessions has not been explored widely in recent literature. By incorporating education on healthy eating alongside a physical activity class, the program provided a small but significant decrease in BMI. This change is reflective of the program success which increased the amount of daily physical activity together with the daily consumption of fruit and vegetable. Important to note, is the average consumption of fruit and vegetable, the level of physical activity and the amount of sedentary behaviour all still failed to meet respective Australian and WHO guidelines [36].

The standard HEAL™ program has previously reported positive health changes for people living in rural areas who are experiencing chronic disease or are at risk of developing chronic disease to positively change their lifestyle [15]. For people experiencing challenges to mental health, in this study, and others, physical activity programs show similar benefits to health and wellbeing. Mental health conditions have also been explored [37–39]. However, the availability of an effective program is of no value if people are not participating in the opportunity, or the program is not able to be sustained [40, 41]. Hutchesson et al identified reduced finances and a lack of volunteers as a barrier to the implementation of a program promoting mental health and well-being in rural areas of South Australia [42]. Stigma associated with mental health and lack of confidentiality in rural communities were identified as issues that may impact the help seeking behaviour of individuals [42]. With effectiveness of the program established, future research is needed to enhance delivery through implementation planning that considers the uniqueness of the clinic and geographic settings being served as well as the people, both delivering (AEP or nurse) and participating in (patients/clients/service users) the HEAL™ing Mental Health program.

This program provides a positive approach for exercise and lifestyle modification in people with mental health conditions and warrant further exploration in a larger sample. Several limitations of the study are important in consideration of these positive findings. The program was designed for those experiencing mental health conditions. Results of the K10 indicated that on

baseline testing, more than half of participants were likely to be well (K-10 score <20). The overall inclusion (sample size) was much lower than expected and precluded planned sub-group analyses, including those by severity of mental health distress. There was no exclusion criteria or data collected for any conditions or comorbidities that could have influenced the outcomes of this study, such as arthritis, chronic obstructive pulmonary disease or diabetes.

It was intended that an AEP lead the program sessions, with eight weeks of attendance by adults experiencing poor mental health. These intentions were not always able to be realised in the setting of real-world delivery. The program was nurse-led (qualified HEAL facilitator) at one site but given the small numbers, it was not possible to compare outcomes between different professional providers. This is important for planning future program delivery to remain flexible with the available workforce in rural settings.

Several participants completed fewer than half of the sessions or started the program and did not complete it. We do not have reasons for participant drop out. Eighteen subjects were subsequently excluded from our analysis, and we were unable to explore the level of attendance required to achieve successful outcomes as we had intended.

The use of self-report instruments to measure physical activity and sedentary time has been found to consistently result in an underestimate of the amount of activity [43] and sedentary time [44]. Objective testing of this parameter might be considered in future studies.

## Conclusions

The provision of a free community-based program was beneficial for those who attended more than 50% of the sessions. The positive results seen in those who did make the necessary commitment to completing the program suggests that the mental health version of the standard HEAL™ program can be used to enhance the engagement and participation of individuals in physical activity.

## Supporting information

**S1 Table. Stage of change and K-10 scores at baseline for included and excluded participants.**
(DOCX)

**S1 Checklist. STROBE statement—Checklist of items that should be included in reports of observational studies.**
(DOCX)

## Acknowledgments

We thank the program delivery teams and participants who completed the HEAL™ ing Mental Health program and data collection, and Exercise & Sports Science Australia (ESSA) for their support of data collection.

## Author Contributions

**Conceptualization:** Annette J. Raynor, Sophia Nimphius, Sally Casson, Caitlin Fox-Harding, Lauren V. Fortington.

**Data curation:** Annette J. Raynor, Lauren V. Fortington.

**Formal analysis:** Sophia Nimphius, Daniel Kadlec.

**Funding acquisition:** Annette J. Raynor, Sophia Nimphius, Lauren V. Fortington.

**Investigation:** Annette J. Raynor, Sally Casson, Caitlin Fox-Harding, Lauren V. Fortington.

**Methodology:** Annette J. Raynor, Sophia Nimphius, Daniel Kadlec, Lauren V. Fortington.

**Project administration:** Annette J. Raynor.

**Supervision:** Annette J. Raynor, Sophia Nimphius, Lauren V. Fortington.

**Visualization:** Sophia Nimphius, Daniel Kadlec.

**Writing – original draft:** Annette J. Raynor, Lauren V. Fortington.

**Writing – review & editing:** Sophia Nimphius, Daniel Kadlec, Sally Casson, Caitlin Fox-Harding.

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
