## [Decision Letter · Decision Letter 0]

23 Oct 2023

PONE-D-23-12328Evaluation of the HEALTMing Mental Health program: a prospective cohort study of short-term changes from a physical activity and lifestyle education program for people with mental health disorders living in rural Australia.PLOS ONE

Dear Dr. Fortington,

Thank you for submitting your manuscript to PLOS ONE. After careful consideration, we feel that it has merit but does not fully meet PLOS ONE’s publication criteria as it currently stands. Therefore, we invite you to submit a revised version of the manuscript that addresses the points raised during the review process.

We look forward to receiving your revised manuscript.

Kind regards,

Syed Afroz Keramat, Ph.D.

Academic Editor

PLOS ONE

Journal Requirements:

3. We note that you have stated that you will provide repository information for your data at acceptance. Should your manuscript be accepted for publication, we will hold it until you provide the relevant accession numbers or DOIs necessary to access your data. If you wish to make changes to your Data Availability statement, please describe these changes in your cover letter and we will update your Data Availability statement to reflect the information you provide

Reviewers' comments:

Reviewer's Responses to Questions

**Comments to the Author**

1. Is the manuscript technically sound, and do the data support the conclusions?

Reviewer #1: Yes

2. Has the statistical analysis been performed appropriately and rigorously? 

Reviewer #1: Yes

3. Have the authors made all data underlying the findings in their manuscript fully available?

Reviewer #1: Yes

4. Is the manuscript presented in an intelligible fashion and written in standard English?

Reviewer #1: Yes

5. Review Comments to the Author

Reviewer #1: Minor comments:

1. The first paragraph of introduction section could have a link sentence at the end regarding the mental health and physical activity/lifestyle factors to indicate that what is coming in the second paragraph. However, the rest part of the introduction is well written.

2. Most of the references at the end of the sentences are after the punctuation (e.g. line # 38,45,51, 56, 58, etc.), which should be before the punctuations.

3. In the the method section, the first mention of ESSA (line 89), CALD (line 94) and AEP (line 102)were acronyms, not abbreviated earlier in the paper.

4. Line 176 to 178: the last part of the sentence is not clear to me, can you please rewrite it?

"This attendance cut-off strengthens the complete data (data present before and after the intervention) as only two participants did not subsequently drop out of program completely"

5. The use of restricted maximum likelihood(REML) estimation in Liner Mixed Model (LMM) is often preferred because it provides more accurate estimates of the variance components and model parameters, especially when the fixed effects and random effects are correlated. REML estimation aims to maximize the likelihood of the random effects while marginalizing over the fixed effects, making it a suitable choice for many practical applications. If this the case to preferring this model in analysing this study data, in the methodology part (from 180 line onward) this should be explained.

6. PLOS authors have the option to publish the peer review history of their article (what does this mean?). If published, this will include your full peer review and any attached files.

Reviewer #1: No

---

## [Author Response · Author response to Decision Letter 0]

12 Jan 2024

Health program: a prospective cohort study of short-term changes from a physical activity and lifestyle education program for people with mental health disorders living in rural Australia”

Enclosed in our response to reviewer document, we outline the changes made as provided by email on 24 October 2023.

---

## [Editor Report · Decision Letter 1]

16 Feb 2024

Evaluation of the HEALTMing Mental Health program: a prospective cohort study of short-term changes from a physical activity and lifestyle education program for people with mental health disorders living in rural Australia.

PONE-D-23-12328R1

Dear Dr. Lauren,

We’re pleased to inform you that your manuscript has been judged scientifically suitable for publication and will be formally accepted for publication once it meets all outstanding technical requirements.

Kind regards,

Syed Afroz Keramat, Ph.D.

Academic Editor

PLOS ONE

Additional Editor Comments (optional):

Congratulations. Great work to be a reference for future works.
---

## [Editor Report · Acceptance letter]

4 Mar 2024

PONE-D-23-12328R1 

PLOS ONE

Dear Dr. Fortington, 

I'm pleased to inform you that your manuscript has been deemed suitable for publication in PLOS ONE. Congratulations! Your manuscript is now being handed over to our production team.

Kind regards, 

on behalf of

Dr. Syed Afroz Keramat 

Academic Editor

PLOS ONE